# Effective and Efficient Batch Normalization Using Few Uncorrelated Data for Statistics' Estimation

## Abstract

Deep Neural Networks (DNNs) thrive in recent years in which Batch Normalization (BN) plays an indispensable role. However, it has been observed that BN is costly due to the reduction operations. In this paper, we propose alleviating the BN's cost by using only a small fraction of data for mean & variance estimation at each iteration. The key challenge to reach this goal is how to achieve a satisfactory balance between normalization effectiveness and execution efficiency. We identify that the effectiveness expects less data correlation while the efficiency expects regular execution pattern. To this end, we propose two categories of approach: sampling or creating few uncorrelated data for statistics' estimation with certain strategy constraints. The former includes "Batch Sampling (BS)" that randomly selects few samples from each batch and "Feature Sampling (FS)" that randomly selects a small patch from each feature map of all samples, and the latter is "Virtual Dataset Normalization (VDN)" that generates few synthetic random samples. Accordingly, multi-way strategies are designed to reduce the data correlation for accurate estimation and optimize the execution pattern for running acceleration in the meantime. All the proposed methods are comprehensively evaluated on various DNN models, where an overall training speedup by up to 21.7% on modern GPUs can be practically achieved without the support of any specialized libraries, and the loss of model accuracy and convergence rate are negligible. Furthermore, our methods demonstrate powerful performance when solving the well-known "micro-batch normalization" problem in the case of tiny batch size.

## 1 Introduction

Recent years, Deep Neural Networks (DNNs) have achieved remarkable success in a wide spectrum of domains such as computer vision (Krizhevsky et al., 2012) and language modeling (Collobert & Weston, 2008). The success of DNNs largely relies on the capability of presentation benefit from the deep structure (Delalleau & Bengio, 2011). However, training a deep network is so difficult to converge that batch normalization (BN) has been proposed to solve it (Ioffe & Szegedy, 2015). BN leverages the statistics (mean & variance) of mini-batches to standardize the activations. It allows the network to go deeper without significant gradient explosion or vanishing (Santurkar et al., 2018; Ioffe & Szegedy, 2015). Moreover, previous work has demonstrated that BN enables the use of higher learning rate and less awareness on the initialization (Ioffe & Szegedy, 2015), as well as produces mutual information across samples (Morcos et al., 2018) or introduces estimation noises (Bjorck et al., 2018) for better generalization.

Despite BN's effectiveness, it is observed that BN introduces considerable training overhead due to the costly reduction operations. The use of BN can lower the overall training speed (mini second per image) by >45% (Wu et al., 2018), especially in deep models. To alleviate this problem, several methods were reported. Range Batch Normalization (RBN) (Banner et al., 2018) accelerated the forward pass by estimating the variance according to the data range of activations within each batch. A similar approach, $L_1$-norm BN (L1BN) (Wu et al., 2018), simplified both the forward and backward passes by replacing the $L_2$-norm variance with its $L_1$-norm version and re-derived the gradients for backpropagation (BP) training. Different from the above two methods, Self-normalization (Klambauer et al., 2017) provided another solution which totally eliminates the need of BN operation with

an elaborate activation function called "scaled exponential linear unit" (SELU). SELU can automatically force the activation towards zero mean and unit variance for better convergence. Nevertheless, all of these methods are not sufficiently effective. The strengths of L1BN & RBN are very limited since GPU has sufficient resources to optimize the execution speed of complex arithmetic operations such as root for the vanilla calculation of $L_2$-norm variance. Since the derivation of SELU is based on the plain convolutional network, currently it cannot handle other modern structures with skip paths like ResNet and DenseNet.

In this paper, we propose mitigating BN's computational cost by just using few data to estimate the mean and variance at each iteration. Whereas, the key challenge of this way lies at how to preserve the normalization effectiveness of the vanilla BN and improve the execution efficiency in the meantime, i.e. balance the effectiveness-efficiency trade-off. We identify that the effectiveness preservation expects less data correlation and the efficiency improvement expects regular execution pattern. This observation motivates us to propose two categories of approach to achieve the goal of effective and efficient BN: sampling or creating few uncorrelated data for statistics' estimation with certain strategy constraints.

Sampling data includes "Batch Sampling (BS)" that randomly selects few samples from each batch and "Feature Sampling (FS)" that randomly selects a small patch from each feature map (FM) of all samples; creating data means "Virtual Dataset Normalization (VDN)" that generates few synthetic random samples, inspired by Salimans et al. (2016). Consequently, multi-way strategies including intra-layer regularity, inter-layer randomness, and static execution graph during each epoch, are designed to reduce the data correlation for accurate estimation and optimize the execution pattern for running acceleration in the meantime. All the proposed approaches with single-use or joint-use are comprehensively evaluated on various DNN models, where the loss of model accuracy and convergence rate is negligible. We practically achieve an overall training speedup by up to 21.7% on modern GPUs. Note that any support of specialized libraries is not needed in our work, which is not like the network pruning (Zhu et al., 2018) or quantization (Hubara et al., 2017) requiring extra library for sparse or low-precision computation, respectively. Most previous acceleration works targeted inference which remained the training inefficient (Wen et al., 2016; Molchanov et al., 2016; Luo et al., 2017; Zhang et al., 2018b; Hu et al., 2018), and the rest works for training acceleration were orthogonal to our approach (Lin et al., 2017; Goyal et al., 2017; You et al., 2017). Additionally, our methods further shows powerful performance when solving the well-known "micro-batch normalization" problem in the case of tiny batch sizes.

In summary, the major contributions of this work are summarized as follows.

- We propose a new way to alleviate BN's computational cost by using few data to estimate the mean and variance, in which we identify that the key challenge is to balance the normalization effectiveness via less data correlation and execution efficiency via regular execution pattern.

- We propose two categories of approach to achieve the above goal: sampling (BS/FS) or creating (VDN) few uncorrelated data for statistics' estimation, in which multi-way strategies are designed to reduce the data correlation for accurate estimation and optimize the execution pattern for running acceleration in the meantime. The approaches can be used alone or jointly.

- Various benchmarks are evaluated, on which up to 21.7% practical acceleration is achieved for overall training on modern GPUs with negligible accuracy loss and without specialized library support.

- Our methods are also extended to the micro-BN problem and achieve advanced performance[1].

In order to make this paper easier for understanding, we present the organization of the whole paper in Fig. 14, Appendix G.

---

[1]Due to the limitation of pages, this part is represented in Appendix F.

## 2 MOTIVATION, OPPORTUNITY, AND CHALLENGE

### 2.1 MOTIVATION: COSTLY BATCH NORMALIZATION & THE BOTTLENECK ANALYSIS

The activations in one layer for normalization can be described by a $d$-dimensional activation feature $\boldsymbol{X} = (x^{(1)}, .., x^{(d)})$, where for each feature we have $x^{(k)} = (x_1^{(k)}, .., x_m^{(k)})$. Note that in convolutional (Conv) layer, $d$ is the number of FMs and $m$ equals to the number of points in each FM across all the samples in one batch; while in fully-connected (FC) layer, $d$ and $m$ are the neuron number and batch size, respectively. BN uses the statistics (mean $E[x^{(k)}]$ & variance $Var[x^{(k)}]$) of the intra-batch data for each feature to normalize activation by

$$\widehat{x}^{(k)} = \frac{x^{(k)} - E[x^{(k)}]}{\sqrt{Var[x^{(k)}] + \epsilon}}, \ y^{(k)} = \gamma^{(k)}\widehat{x}^{(k)} + \beta^{(k)} \tag{1}$$

where $\gamma^{(k)}$ & $\beta^{(k)}$ are trainable parameters introduced to recover the representation capability, $\epsilon$ is a small constant to avoid numerical error, and $E[x^{(k)}]$ & $Var[x^{(k)}]$ can be calculated by

$$E[x^{(k)}] = \frac{1}{m}\sum_{j=1}^{m}x_j^{(k)}, Var[x^{(k)}] = \frac{1}{m}\sum_{j=1}^{m}(x_j^{(k)} - E[x^{(k)}])^2. \tag{2}$$

The detailed operations of a BN layer in the backward pass can be found in Appendix C.

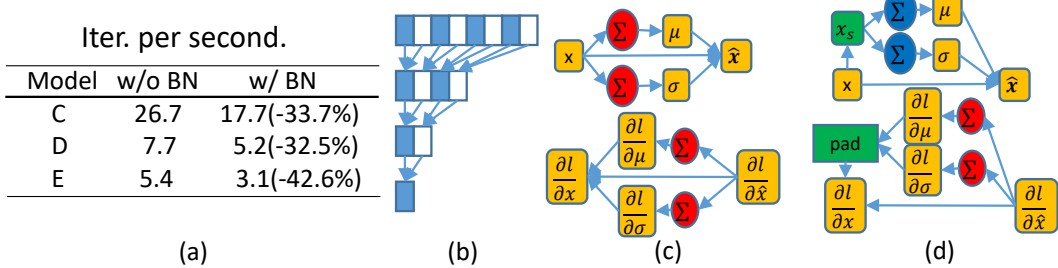

| Iter. per second. | | |
|---|---|---|
| Model | w/o BN | w/ BN |
| C | 26.7 | 17.7(-33.7%) |
| D | 7.7 | 5.2(-32.5%) |
| E | 5.4 | 3.1(-42.6%) |

(a)       (b)       (c)       (d)

Figure 1: Illustration of BN's cost, operations, bottleneck, and opportunity: (a) iterations per second for training various models with or without BN, Model C,D,E are defined in Table 2; (b) usual optimization of the reduction operation using adder tree; (c) the computational graph of BN in the forward pass (upper) and backward pass (lower); (d) the computation graph of BN using few data for statistics' estimation in forward pass (upper) and backward pass (lower). $x$ is neuronal activations, $\mu$ and $\sigma$ denote the mean and standard deviation of $x$ within one batch, respectively, and $\sum$ is the summation operation.

From Fig. 1(a), we can see that adding BN will significantly slow down the training speed (iterations per second) by 32%-43% on ImageNet. The reason why BN is costly is that it contains several "reduction operations", i.e. $\sum_{j=1}^{m}$. We offer more thorough data analysis in Appendix E. If the reduction operations are not optimized, it's computational complexity should be $O(m)$. With the optimized parallel algorithm proposed in (Che et al., 2008), the reduction operation is transformed to cascaded adders of depth of $log(m)$ as shown in Fig. 1(b). However, the computational cost is still high since we usually have $m$ larger than one million. As shown in Fig. 1(c), the red "$\sum$"s represent operations that contain summations, which cause the BN inefficiency.

### 2.2 OPPORTUNITY: USING FEW DATA FOR STATISTICS' ESTIMATION

Motivated by the above analysis, decreasing the effective value of $m$ at each time for statistics estimation seems a promising way to reduce the BN cost for achieving acceleration. To this end, we propose using few data to estimate the mean and variance at each iteration. For example, if $m$ changes to a much smaller value of $s$, equation (2) can be modified as

$$E[x^{(k)}] \approx E[x_s^{(k)}] = \frac{1}{s}\sum_{j=1}^{s}x_j^{(k)}, \ Var[x^{(k)}] \approx Var[x_s^{(k)}] = \frac{1}{s}\sum_{j=1}^{s}(x_j^{(k)} - E[x_s^{(k)}])^2 \tag{3}$$

where $x_s^{(k)}$ denotes the small fraction of data, $s$ is the actual number of data points, and we usually have $s \ll m$. Here we denote $s/m$ as **Sampling Ratio** (it includes both the cases of sampling and creating few data in Section 3.1 and 3.2, respectively). Since the reduction operations in the backward pass can be parallelized whereas in the forward pass, the variance can not be calculated until mean is provided (which makes it nearly twice as slow as backward pass), we just use few data in the forward pass. The computational graph of BN using few data is illustrated in Fig. 1(d). The key is how to estimate $E[x^{(k)}]$ & $Var[x^{(k)}]$ for each neuron or FM within one batch with much fewer data. The influence on the backward pass is discussed in Appendix C.

### 2.3 CHALLENGE: EFFECTIVENESS-EFFICIENCY BALANCE

Although using few data can reduce the BN's cost dramatically, it will meet an intractable challenge: how to simultaneously preserve normalization effectiveness and improve the execution efficiency. On one side, using few data to estimate the statistics will increase the estimation error. By regarding the layers with high estimation error as unnormalized ones, we did contrast test in Fig. 2. The mean & variance will be scaled up exponentially as network deepens, which causes the degradation of BN's effectiveness. This degradation can be recovered from two aspects.

- **Intra-layer**. For the reason that the estimation error is not only determined by the amount of data but also the correlation between them, we can sample less correlated data within each layer to improve the estimation accuracy.

- **Inter-layer**. As depicted in by Fig. 2, the intermittent BN configuration (i.e. discontinuously adding BN in different layers) can also prevent the statistics scaling up across layers. This motivates us that as long as layers with high estimation error are discontinuous, the statistics shift can still be constrained to a smaller range. Therefore, to reduce the correlation between estimation errors in different layers can also be beneficial to improve the accuracy of the entire model, which can be achieved by sampling less correlated data between layers.

On the other side, less data correlation indicates more randomness which usually causes irregular memory access irregular degrading the running efficiency. In this paper, we recognize that the overhead of sampling can be well reduced by using regular and static execution patterns, which is demonstrated with ablation study at Fig. 9 (c), Sec. 4.2.

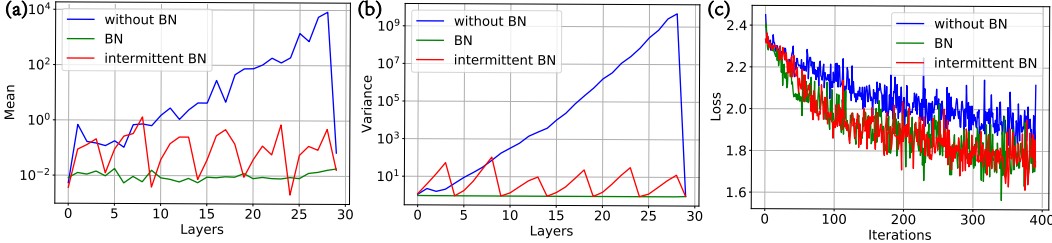

Figure 2: The influence of BN configuration on deep models: (a) & (b) average mean & variance of each layer during the first training epoch; (c) loss curve during the first training epoch. Here we train a 30-layer CNN on CIFAR-10. For "without BN" curve, BN is only applied to the last Conv layer; For "BN" curve, BN is applied to all Conv layers; For "intermittent BN" curve, BN is applied to layer $5k + 4$, where $k \in \{0, 1, 2, 3, 4, 5\}$.

In a nutshell, careful designs are needed to balance the normalization effectiveness via less data correlation and the execution efficiency via regular execution pattern. Only in this way, it is possible to achieve practical acceleration with little accuracy loss, which is our major target in this work. Based on the above analysis, we summarize the design considerations as follows.

- Using few data for statistics' estimation can effectively reduce the computational cost of BN operations. Whereas, the effectiveness-efficiency trade-off should be well balanced.

- Less data correlation is promising to reduce the estimation error and then guarantees the normalization effectiveness.

- More regular execution pattern is expected for efficient running on practical platforms.

## 3 APPROACHES

To reach the aforementioned goal, we propose two categories of approach in this section: sampling or creating few uncorrelated data for statistics' estimation. Furthermore, multi-way strategies to balance the data correlation and execution regularity.

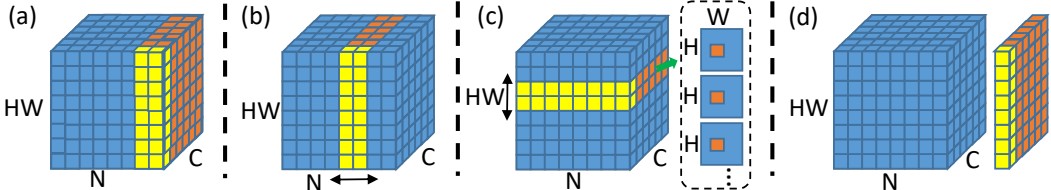

Figure 3: Illustration of approaches: (a) Naive Sampling; (b) Batch Sampling; (c) Feature Sampling; (d) Virtual Dataset Normalization. 'H' and 'W' are the height and width of FMs, respectively, 'C' is the number of FMs for current layer, and 'N' denotes the number of samples in current batch. The orange and yellow rectangles in (a)-(c) represent the sampled data and those in (d) are the created virtual sample(s). The yellow data are used to estimate the statistics for current FM.

### 3.1 SAMPLING UNCORRELATED DATA: BATCH SAMPLING AND FEATURE SAMPLING

**Uncorrelated Sampling**. Here "sampling" means to sample a small fraction of data from the activations at each layer for statistics' estimation. As discussed in the previous section, mining uncorrelated data within and between layers is critical to the success of sampling-based BN. However, the correlation property of activations in deep networks is complex and may vary across network structures and datasets. Instead, we apply a **hypothesis-testing** approach in this work.

We first make two empirical assumptions:

- **Hypothesis 1**. Within each layer, **data belonging to the different samples** are more likely to be uncorrelated than those within the same sample.

- **Hypothesis 2**. Between layers, **data belonging to different locations and different samples** are less likely to be correlated. Here "location" means coordinate within FMs.

These two assumptions are based on the basic nature of real-world data and the networks, thus they are likely to hold in most situations. They are further evaluated through experiments in Section 4.1.

Based on above hypotheses, we propose two uncorrelated-sampling strategies: **Batch Sampling (BS)** and **Feature Sampling (FS)**. The detailed Algorithms can be found in Alg. 1 and 2, respectively, Appendix A.

- **BS** (Fig. 3(b)) randomly selects few samples from each batch for statistics' estimation. To reduce the inter-layer data correlation, it selects different samples across layers following Hypothesis 2.

- **FS** (Fig. 3(c)) randomly selects a small patch from each FM of all samples for statistics' estimation. Since the sampled data come from all the samples thus it has lower correlation within each layer following Hypothesis 1. Furthermore, it samples different patch locations across layers to reduce the inter-layer data correlation following Hypothesis 2.

A Naive Sampling (NS) is additionally proposed as a comparison baseline, as shown in Fig. 3(a). NS is similar to BS while the sampling index is fixed across layers, i.e. consistently samples first few samples within each batch.

**Regular and Static Sampling**. In order to achieve practical acceleration on GPU, we expect more regular sampling pattern and more static sampling index. Therefore, to balance the estimation effectiveness and execution efficiency, we carefully design the following sampling rules: (1) In BS, the selected samples are continuous and the sample index for different channels are shared, while they are independent across layers; (2) In FS, the patch shape is rectangular and the patch location is shared by different channels and samples within each layer but variable as layer changes. Furthermore, all the random indexes are updated only once for each epoch, which guarantees a static computational graph during the entire epoch.

### 3.2 CREATING UNCORRELATED DATA: VIRTUAL DATASET NORMALIZATION

Instead of sampling uncorrelated data, another plausible solution is to directly create uncorrelated data for statistics' estimation. We propose Virtual Dataset Normalization (VDN)[2] to implement it, as illustrated in Fig. 3(d). VDN is realized with the following three steps: (1) calculating the statistics of the whole training dataset offline; (2) generating $s$ virtual samples[3] at each iteration to concatenate with the original real inputs as the final network inputs. (3) using data from only virtual samples at each layer for statistics' estimation. Due to the independent property of the synthesized data, they are more uncorrelated than real samples thus VDN can produce much more accurate estimation. The detailed implementation algorithm can be found in Alg. 3, Appendix A.

### 3.3 SINGLE USE OR JOINT USE

The sampling approach and the creating approach can be used either in a single way or in a joint way. Besides the single use, a joint use can be described as

$$
\begin{aligned}
E[x^{(k)}] &= \beta E[x_s^{(k)}] + (1-\beta)E[x_v^{(k)}] \\
Var[x^{(k)}] &= \beta Var[x_s^{(k)}] + (1-\beta)Var[x_v^{(k)}] + \beta(1-\beta)(E[x_s^{(k)}] - E[x_v^{(k)}])^2
\end{aligned}
\tag{4}
$$

where $x_s$ denotes the sampled real data while $x_v$ represents the created virtual data. $\beta$ is a controlling variable, which indicates how large the sampled data occupy the whole data for statistics' estimation within each batch: (1) when $\beta = 0$ or 1, the statistics come from single approach (VDN or any sampling strategy); (2) when $\beta \in (0,\ 1)$, the final statistics are a joint value as shown in equation 4.

### 3.4 COMPARISON BETWEEN DIFFERENT USING WAYS

A comparison between different using ways is presented in Table 1, where "d.s." denotes "different samples"; "d.l." stands for "different locations", and "g.i." indicates "generated independent data". Compared to NS, BS reduces the inter-layer correlation via selecting different samples across layers; FS reduces both the intra-layer and inter-layer correlation via using data from all samples within each layer and selecting different locations across layers, respectively. Though VDN has a similar inter-layer correlation with NS, it slims the intra-layer correlation with strong data independence. A combination of BS/FS and VDN can inherit the strength of different approaches, thus achieve much lower accuracy loss.

Table 1: Comparison between different using ways.

|  | NS | BS | FS | VDN | BS/FS + VDN |
|---|---|---|---|---|---|
| Inter-layer Correlation | × | d.s. | d.l. | × | d.l./d.c. |
| Intra-layer Correlation | × | × | d.s. | g.i. | ×/d.s. + g.i. |

## 4 EXPERIMENTS

**Experimental Setup**. All of our proposed approaches are validated on image classification task using CIFAR-10, CIFAR-100 and ImageNet datasets from two perspectives: (1) effectiveness evaluation and (2) efficiency execution. To demon-

Table 2: Model configuration.

| Model | Dataset | Network | Samples/GPU | GPU |
|---|---|---|---|---|
| A | CIFAR-10 | ResNet-20 | 128 | Titan Xp ×1 |
| B | CIFAR-10/100 | ResNet-56 | 128 | Titan Xp ×1 |
| C | ImageNet | ResNet-18 | 32 | Tesla V100 ×1 |
| D | ImageNet | ResNet-18 | 128 | Tesla V100 ×2 |
| E | ImageNet | ResNet-50 | 64 | Tesla V100 ×4 |
| F | ImageNet | DenseNet-121 | 64 | Tesla V100 ×3 |

strate the scalability and generality of our approaches on deep networks, we select ResNet-56 on CIFAR-10 & CIFAR-100 and select ResNet-18 and DenseNet-121 on ImageNet[4]. The model

---

[2]Virtual: synthesized samples rather than real ones are used for statistics' estimation; Dataset: the virtual samples are generated following the dataset's distribution.

[3]Virtual samples: random tensors wherein each element simply follows $N(\mu, \sigma)$ where $\mu$ and $\sigma$ are the statistics of the whole training dataset calculated offline in step (1).

[4]On ImageNet, most experiments are conducted on ResNet-18 due to the faster training time, and only joint approaches are demonstrated on on DenseNet-121. The accuracy and speedup results are on DenseNet-121 are enough to evidence the effectiveness and efficiency of our approaches for deep models.

configuration can be found in Table 2. The means and variances for BN are locally calculated in each GPU without inter-GPU synchronization as usual. We denote our approaches as the format of "Approach-Sampled_size/Original_size-sampling_ratio(%)". For instance, if we assume batch size is 128, "BS-4/128-3.1%" denotes only 4 samples are sampled in BS and the sampling ratio equals to $\frac{4}{128} = 3.1\%$. Similarly, "FS-1/32-3.1%" implies a $\frac{1}{32} = 3.1\%$ patch is sampled from each FM, and "VDN-1/128-0.8%" indicates only one virtual sample is added. The traditional BN is denoted as "BN-128/128-100.0%". Other experimental configurations can be found in Appendix B.

## 4.1 Effectiveness Evaluation

**Convergence Analysis**. Fig. 4 shows the top-1 validation accuracy and confidential interval of ResNet-56 on CIFAR-10 and CIFAR-100. On one side, all of our approaches can well approximate the accuracy of normal BN when the sampling ratio is larger than $2\%$, which evidence their effectiveness. On the other side, all the proposed approaches perform better than the NS baseline. In particular, **FS** performs best, which is robust to the sampling ratio with negligible accuracy loss (e.g. at sampling ratio=1.6%, the accuracy degradation is **-0.087%** on CIFAR-10 and **+0.396%** on CIFAR-100).

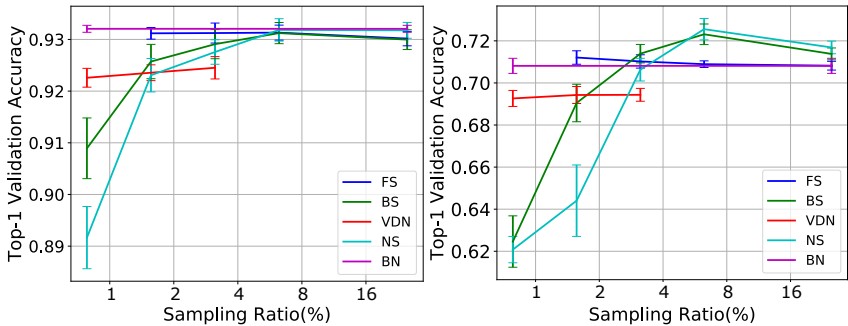

Figure 4: Top-1 validation accuracy of ResNet-56 on CIFAR-10 (left) & CIFAR-100 (right).

**VDN** outperforms BS and NS with a large margin in extremely small sampling ratio (e.g. 0.8%), whereas the increase of virtual batch size leads to little improvement on accuracy. **BS** is constantly better than NS. Furthermore, an interesting observation is that the BN sampling could even achieve better accuracy sometimes, such as NS-8/128(**72.6±1.5%**), BS-8/128(**72.3±1.5%**), and FS-1/64(**71.2±0.96%**) against the baseline (**70.8±1%**) on CIFAR-100. Fig.5 further shows the training curves of ResNet-56 on CIFAR-10 under different approaches. It reveals that FS and VDN would not harm the convergence rate, while BS and NS begin to degrade the convergence when the sampling ratio is smaller than 1.6% and 3.1%, respectively.

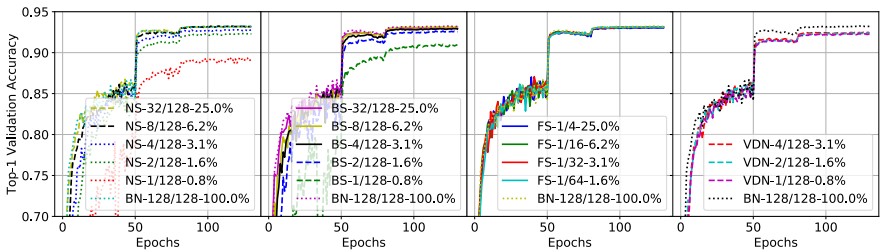

Figure 5: Training curves of ResNet56 on CIFAR-10.

Table 3 shows the top-1 validation error on ImageNet under different approaches. With the same sampling ratio, all the proposed approaches significantly outperform NS, and **FS** surpasses VDN and BS. Under the extreme sampling ratio of 0.78%, NS and BS don't converge. Due to the limitation of FM size, the smallest sampling ratio of **FS** is 1.6%, which has only -0.5% accuracy loss. **VDN** can still achieve relatively low accuracy loss (1.4%) even if the sampling ratio decreases to 0.78%. This implies that VDN is effective for normalization. Moreover, by combining FS-1/64 and VDN-2/128,

Table 3: Top-1 validation error on ImageNet.

| Model | Approach | Sampling Ratio | Top-1 Error(%) | Accuracy Loss(%) |
|---|---|---|---|---|
| **ResNet-18** (256, 128) | BN | 128/128(100%) | 29.8 | baseline |
| | NS | 1/128(0.78%) | N.A. | N.A. |
| | | 4/128(3.1%) | 35.2 | -5.42 |
| | BS | 1/128(0.78%) | N.A. | N.A. |
| | | 4/128(3.1%) | 31.7 | -1.9 |
| | VDN | 1/128(0.78%) | 31.2 | -1.4 |
| | | 2/128(1.6%) | 30.8 | -1.0 |
| | FS | 1/64(1.6%) | 30.3 | -0.5 |
| | | 1/32(3.1%) | 30.4 | -0.6 |
| | **FS+VDN** | 4/128(3.1%) | 30.0 | **-0.2** |
| **DenseNet-121** (192, 64) | BN | 64/64(100%) | 26.1 | baseline |
| | NS | 3/64(4.7%) | N.A. | N.A. |
| | BS+VDN | (1+2)/64(4.7%) | 28.5 | -2.4 |
| | **FS+VDN** | (1+2)/64(4.7%) | 26.7 | **-0.6** |

we get the lowest accuracy loss (-0.2%). This further indicates that VDN can be combined with other sampling strategies to achieve better results. Since training DenseNet-121 is time-consuming, we just report the results with **FS/BS-VDN joint** use. Although DenseNet-121 is more challenging than ResNet-18 due to the much deeper structure, the "FS-1/64 + VDN-2/64" can still achieve very low accuracy loss (-0.6%). "BS-1/64 + VDN-2/64" has a little higher accuracy loss, whereas it still achieves better result than NS. In fact, we observed gradient explosion if we just use VDN on very deep network (i.e. DenseNet-121), which can be conquered through jointly applying VDN and other proposed sampling approach (e.g. FS+VDN). Fig. 6 illustrates the training curves for better visualization of the convergence. Except for the BS with extremely small sampling ratio (0.8%) and NS, other approaches and configurations can achieve satisfactory convergence. Here, we further evaluate the fully random sampling (FRS) strategy, which samples completely random points in both the batch and FM dimensions. We can see that FRS is less stable compared with our proposed approaches (except the NS baseline) and achieves much lower accuracy. One possible reason is that under low sampling ratio, the sampled data may occasionally fall into the worse points, which lead to inaccurate estimation of the statistics.

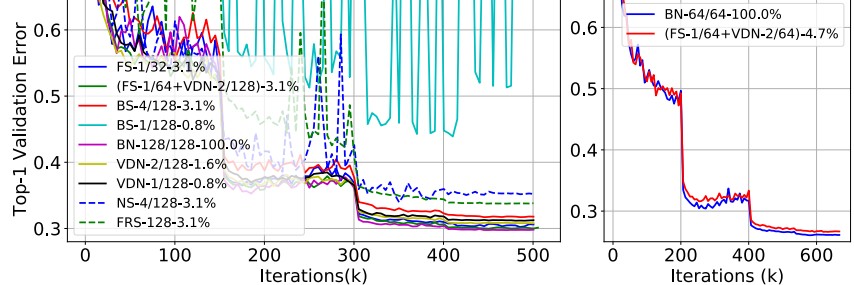

Figure 6: Training curves of ResNet18 (left) and DenseNet121 (right) on ImageNet.

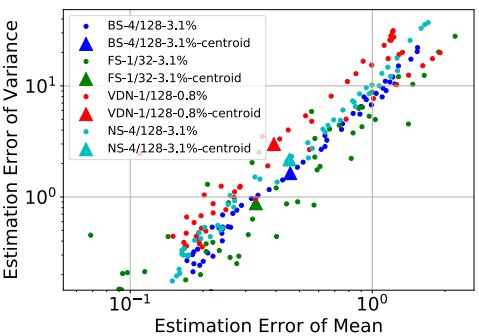

Figure 7: Estimation error distribution.

**Correlation Analysis**. In this section, we bring more empirical analysis to the data correlation that affects the error of statistical estimation. Here we denote the estimation errors at $l^{th}$ layer as $E_\mu^{(l)} = ||\mu_s^{(l)} - \mu^{(l)}||_2$ and $E_\sigma^{(l)} = ||\sigma_s^{(l)} - \sigma^{(l)}||_2$, where $\mu_s^{(l)}$ & $\sigma_s^{(l)}$ are the estimated mean & variance from the sampled data (including the created data in VDN) while $\mu^{(l)}$ & $\sigma^{(l)}$ are the ground truth from the vanilla BN for the whole batch.

The analysis is conducted on ResNet-56 over CIFAR-10. The estimation errors of all layers are recorded throughout the first training epoch. Fig. 7 and Fig. 8 present the distribution of estimation errors for all layers and the inter-layer correlation between estimation errors, respectively. In Fig. 7, FS demonstrates the least estimation error within each layer which implies its better convergence. The estimation error of VDN seems similar to BS and NS here, but we should note that it uses much lower sampling ratio of 0.8% compared to others

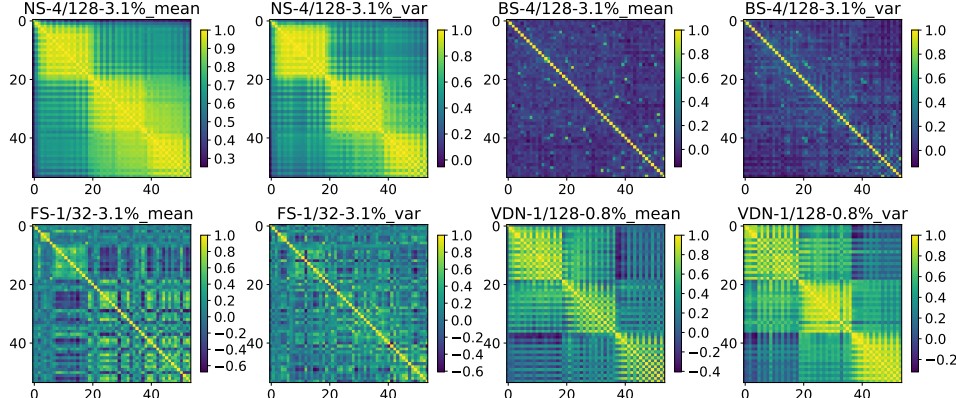

Figure 8: Inter-layer correlation between estimation errors. Both x & y axes denote layer index.

of 3.1%. From Fig. 8, it can be seen that BS presents obviously less inter-layer correlation than NS, which is consistent with previous experimental results that BS can converge better than NS even though they have similar estimation error as shown in Fig. 7. For FS and VDN, although it looks like they present averagely higher correlations, there exist negative corrections which effectively improve the model accuracy. Moreover, FS produces better accuracy than NS and BS since its selected data come from all the samples with less correlation.

## 4.2 EFFICIENCY EVALUATION

After the normalization effectiveness evaluation, we will evaluate the execution efficiency which is the primary motivation. Fig. 9 shows the BN speedup during training and overall training improvement.

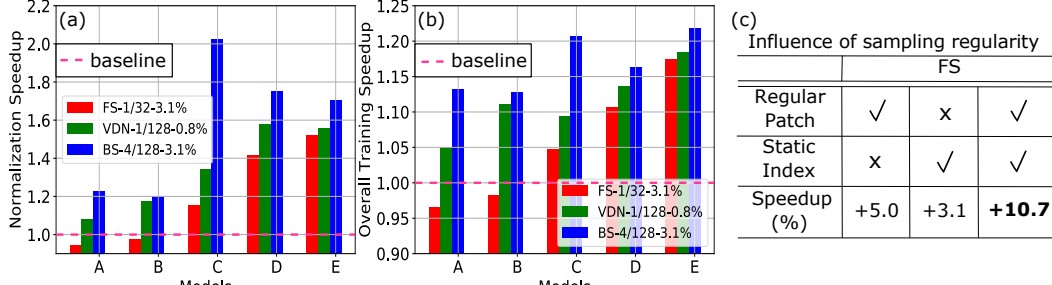

Figure 9: Efficiency evaluation: (a) BN speedup; (b) overall training speedup; (c) influence of sampling regularity (ImageNet, ResNet-18, FS-1/32-3.1%) on overall training speed.

In general, BS can gain higher acceleration ratio because it doesn't incur the fine-grained sampling within FMs like in FS and it doesn't require the additional calculation and concatenation of the virtual samples like in VDN. As for FS, it fails to achieve speedup on CIFAR-10 due to the small image size that makes the reduction of operations unable to cover the sampling overhead. The proposed approaches can obtain up to 2x BN acceleration and 21.8% overall training acceleration.

Table 4 gives more results on ResNet-18 using single approach and DenseNet-121 using joint approach. On ResNet-18, we perform much faster training compared

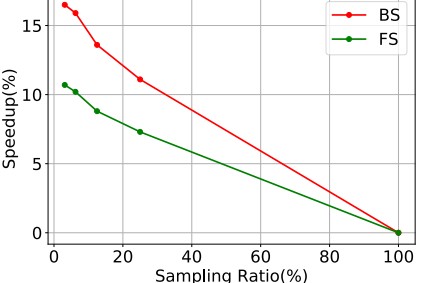

Figure 10: Sampling ratio v.s. speedup.

with two recent methods for BN simplification (Wu et al., 2018; Banner et al., 2018). On ResNet-18 we can achieve up to 16.5% overall training speedup under BS; on very deep networks with more BN layers, such as DenseNet-121, the speedup is more significant that reaches 23.8% under "BS+VDN" joint approach. "FS+VDN" is a little bit slower than "BS+VDN" since the latter one has a more regular execution pattern as aforementioned. Nonetheless, on a very deep model, we still recommend the "FS+VDN" version because it can preserve the accuracy better. The relationship be-

tween sampling ratio and overall training speedup is represented in Fig. 10 which illustrates that 1) BS & FS can still achieve considerable speedup with a moderate sampling ratio; 2) BS can achieve more significant acceleration than FS, for its more regular execution pattern. It's worth noting that, our training speedup is practically obtained on modern GPUs without the support of specialized library that makes it easy-to-use. Fig. 9 (c) reveals that the regular execution pattern can significantly help us achieve practical speedup.

Table 4: Lateral comparison of overall training speed.

| Approach | Model | Sampling Ratio | Iter. per Second | Speedup(%) |
|---|---|---|---|---|
| **ResNet-18** (256, 128) | BN | 128/128 | 5.21 | baseline |
| | L1BN (Wu et al., 2018) | 128/128 | 5.23 | +0.38 |
| | RBN (Banner et al., 2018) | 128/128 | 5.30 | +1.73 |
| | FS | 1/4(25%) | 5.59 | +7.3 |
| | | 1/8(12.5%) | 5.67 | +8.8 |
| | | 1/16(6.25%) | 5.74 | +10.2 |
| | | 1/32(3.1%) | 5.77 | +10.7 |
| | VDN | 1/128(0.78%) | 5.93 | +13.8 |
| | BS | 32/128(25%) | 5.79 | +11.1 |
| | | 16/128(12.5%) | 5.92 | +13.6 |
| | | 8/128(6.25%) | 6.04 | +15.9 |
| | | 4/128(3.1%) | 6.07 | +16.5 |
| **DenseNet-121** (192, 64) | BN | 64/64 | 2.44 | baseline |
| | BS+VDN | (1+2)/64(4.7%) | 3.02 | +23.8 |
| | FS+VDN | (1+2)/64(4.7%) | 2.97 | +21.7 |

## 5  RELATED WORK

**BN** has been applied in most state-of-art DNN models (He et al., 2016; Szegedy et al., 2017) since it was proposed. As aforementioned, BN standardizes the activation distribution to reduce the internal covariate shift. Models with BN have been demonstrated to converge faster and generalize better (Ioffe & Szegedy, 2015; Morcos et al., 2018). Recently, a model called Decorrelated Batch Normalization (DBN) was introduced which not only standardizes but also whitens the activations with ZCA whitening (Huang et al., 2018). Although DBN further improves the normalization performance, it introduces significant extra computational cost.

**Simplifying BN** has been proposed to reduce BN's computational complexity. For example, L1BN (Wu et al., 2018) and RBN (Banner et al., 2018) replace the original $L_2$-norm variance with an $L_1$-norm version and the range of activation values, respectively. From another perspective, Self-normalization uses the customized activation function (SELU) to automatically shift activation's distribution (Klambauer et al., 2017). However, as mentioned in Introduction, all of these methods fail to obtain a satisfactory balance between the effective normalization and computational cost, especially on large-scale modern models and datasets. Our work attempts to address this issue.

## 6  CONCLUSION

Motivated by the importance but high cost of BN layer, we propose using few data to estimate the mean and variance for training acceleration. The key challenge towards this goal is how to balance the normalization effectiveness with much less data for statistics' estimation and the execution efficiency with irregular memory access. To this end, we propose two categories of approach: sampling (BS/FS) or creating (VDN) few uncorrelated data, which can be used alone or jointly. Specifically, BS randomly selects few samples from each batch, FS randomly selects a small patch from each FM of all samples, and VDN generates few synthetic random samples. Then, multi-way strategies including intra-layer regularity, inter-layer randomness, and static execution graph are designed to reduce the data correlation and optimize the execution pattern in the meantime. Comprehensive experiments evidence that the proposed approaches can achieve up to 21.7% overall training acceleration with negligible accuracy loss. In addition, VDN can also be applied to the micro-BN scenario with advanced performance. This paper preliminary proves the effectiveness and efficiency of BN using few data for statistics' estimation. We emphasize that the training speedup is practically achieved on modern GPUs, and we do not need any support of specialized libraries making it easy-to-use. Developing specialized kernel optimization deserves further investigation for more aggressive execution benefits.

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

## APPENDIX A   IMPLEMENTATION ALGORITHMS

**Notations**. We use the Conv layer for illustration, which occupies the major part of most modern networks (He et al., 2016; Huang et al., 2016). The batched features can be viewed as a 4D tensor. We use "$E_{0,1,2}$" and "$Var_{0,1,2}$" to represent the operations that calculate the means and variances, respectively, where "0, 1, 2" denotes the dimensions for reduction.

---

**Algorithm 1:** NS/BS Algorithm

---

**Data:** input batch at layer $l$: $\mathbf{B}_l \in \mathbb{R}^{N \times H_l \times W_l \times C_l}$, $l \in$ all layers; sampling size: $n_s \in (0, N]$
**Result:** estimation of $E[\mathbf{B}_l]$ & $Var[\mathbf{B}_l]$, $l \in$ all layers
**begin**
  **for** $ep \in$ *all epochs* **do**
    **for** $l \in$ *all layers* **do**
      if BS: $begin_l = randint(0, N - n_s)$; else NS: $begin_l = 0$
    **for** $it \in$ *all iterations* **do**
      **for** $l \in$ *all layers* **do**
        $\mathbf{B}_s = \mathbf{B}_l[begin_l : begin_l + n_s - 1, :, :, :]$
        $E[\mathbf{B}_l] = E_{0,1,2}[\mathbf{B}_s], Var[\mathbf{B}_l] = Var_{0,1,2}[\mathbf{B}_s]$

---

**Algorithm 2:** FS Algorithm

---

**Data:** input batch at layer $l$: $\mathbf{B}_l \in \mathbb{R}^{N \times H_l \times W_l \times C_l}$, $l \in$ all layers; sampling size: $h_s^{(l)} \in (0, H_l]$ & $w_s^{(l)} \in (0, W_l]$
**Result:** estimation of $E[\mathbf{B}_l]$ & $Var[\mathbf{B}_l]$, $l \in$ all layers
**begin**
  **for** $ep \in$ *all epochs* **do**
    **for** $l \in$ *all layers* **do**
      $begin_h^{(l)} = randint(0, H_l - h_s^{(l)}), begin_w^{(l)} = randint(0, W_l - w_s^{(l)})$
    **for** $it \in$ *all iterations* **do**
      **for** $l \in$ *all layers* **do**
        $\mathbf{B}_s = \mathbf{B}[:, begin_h^{(l)} : begin_h^{(l)} + h_s^{(l)} - 1, begin_w^{(l)} : begin_w^{(l)} + w_s^{(l)} - 1, :]$
        $E[\mathbf{B}_l] = E_{0,1,2}[\mathbf{B}_s], Var[\mathbf{B}_l] = Var_{0,1,2}[\mathbf{B}_s]$

---

**Algorithm 3:** VDN Algorithm

---

**Data:** Dataset: $\mathbf{D} \in \mathbb{R}^{N_D \times H_0 \times W_0 \times C_0}$; input batch: $\mathbf{B}_l \in \mathbb{R}^{N \times H_l \times W_l \times C_l}$; number of virtual samples: $n_v$
**Result:** estimation of $E[\mathbf{B}_l]$ & $Var[\mathbf{B}_l]$ at layer $l$, $l \in$ all layers
**begin**
  Calculate $\mu = E_{0,1,2}[\mathbf{D}]$ & $\sigma^2 = Var_{0,1,2}[\mathbf{D}]$ offline
  **for** $ep \in$ *all epochs* **do**
    **for** $it \in$ *all iterations* **do**
      Create virtual samples $\mathbf{V} \in \mathbb{R}^{n_v \times H_0 \times W_0 \times C_0}$, $\mathbf{V} \sim N(\mu, \sigma)$
      **for** $l \in$ *all layers* **do**
        if $l = 0$: $\mathbf{B}_l = [\mathbf{V}, \mathbf{B}_l]$, then feed $\mathbf{B}_l$ into the network
        $\mathbf{B}_s = \mathbf{B}_l[0 : n_v - 1, :, :, :]$
        $E[\mathbf{B}_l] = E_{0,1,2}[\mathbf{B}_s], Var[\mathbf{B}_l] = Var_{0,1,2}[\mathbf{B}_s]$

---

## APPENDIX B   EXPERIMENTAL CONFIGURATION

All the experiments on CIFAR-10 & CIFAR-100 are conducted on a single Nvidia Titan Xp GPU. We use a weight decay of 0.0002 for all weight layers and all models are trained by 130 epochs. The

initial learning rate is set to 0.1 and it is decreased by 10x at 50, 80, 110 epoch. During training, we adopt the "random flip left & right" and all the input images are randomly cropped to $32 \times 32$. Each model is trained from scratch for 5 times in order to reduce random variation.

For ImageNet, We use 2 Nvidia Tesla V100 GPUs on DGX station for ResNet-18 and 3 for DenseNet-121. We use a weight decay of 0.0001 for all weight layers and all models are trained by 100 epochs. The initial learning rate is set to $0.1/256 \times$ "$Gradient\ batch\ size$" and we decrease the learning rate by 10x at 30, 60, 80, 90 epoch. During training, all the input images are augmented by random flipping and cropped to $224 \times 224$. We evaluate the top-1 validation error on the validation set using the centered crop of each image. To reduce random variation, we use the average of last 5 epochs to represent its final error rate. Besides, Winograd (Lavin & Gray, 2016) is applied in all models to speedup training.

## APPENDIX C    ANALYSIS OF SPEEDUP

**Compute Cycle and Memory Access**. Our proposed approaches can effectively speedup forward pass. Under the condition that we use $s \ll m$ data to estimation the statistics for each feature, the total accumulation operations are significantly reduced from $m - 1$ to $s - 1$. If using the adder tree optimization illustrated in Section 2.1, the tree's depth can be reduced from $log(m)$ to $log(s)$. Thus, the theoretical compute speedup for the forward pass can reach $log_s(m)$ times. For instance, if the FM size is $56 \times 56$ with batch size of 128 ($m = 56 \times 56 \times 128$), under sampling ratio of 1/32, the compute speedup will be 36.7%. The total memory access is reduced by $m/s$ times. For example, when the sampling ratio is 1/32, only 3.1% data need to be visited. This also contributes a considerable part in the overall speedup.

**Speedup in the Backward Pass**. The BN operations in the forward pass have been shown in equation (1)-(3). Based on the derivative chain rule, we can get the corresponding operations in the backward pass as follows

$$\frac{\partial l}{\partial \widehat{x}_i^{(k)}} = \frac{\partial l}{\partial y_i(k)} \cdot \gamma^{(k)}, \quad \frac{\partial l}{\partial E[x^{(k)}]} = \left(\sum_{j=1}^m \frac{\partial l}{\partial \widehat{x}_j^{(k)}} \cdot \frac{-1}{\sqrt{Var[x^{(k)}] + \epsilon}}\right)$$

$$\frac{\partial l}{\partial Var[x^{(k)}]} = \sum_{j=1}^m \frac{\partial l}{\partial \widehat{x}_j^{(k)}} \cdot (x_j - E[x^{(k)}]) \cdot \frac{-1}{2}(Var[x^{(k)}] + \epsilon)^{-3/2}$$

$$\frac{\partial l}{\partial x_i^{(k)}} = \frac{\partial l}{\partial \widehat{x}_i^{(k)}} \cdot \frac{1}{\sqrt{Var[x^{(k)}] + \epsilon}} + \frac{\partial l}{\partial E[x^{(k)}]} \cdot \frac{1}{m} + \frac{\partial l}{\partial Var[x^{(k)}]} \cdot \frac{2(x_i^{(k)} - E[x^{(k)}])}{m} \quad (5)$$

$$\frac{\partial l}{\partial \gamma^{(k)}} = \sum_{i=1}^m \frac{\partial l}{\partial y_i^{(k)}} \cdot \widehat{x}_i^{(k)}, \quad \frac{\partial l}{\partial \beta^{(k)}} = \sum_{i=1}^m \frac{\partial l}{\partial y_i^{(k)}}$$

Take the BN sampling as an example, the calculation of $\frac{\partial l}{\partial x_i^{(k)}}$ can be modified as

$$\frac{\partial l}{\partial x_i^{(k)}} = \begin{cases} \frac{\partial l}{\partial \widehat{x}_i^{(k)}} \cdot \frac{1}{\sqrt{Var[x^{(k)}] + \epsilon}} + \frac{\partial l}{\partial E[x^{(k)}]} \cdot \frac{1}{s} + \frac{\partial l}{\partial Var[x^{(k)}]} \cdot \frac{2(x_i^{(k)} - E[x^{(k)}])}{s}, & \text{if } x_i \in S \\ \frac{\partial l}{\partial \widehat{x}_i^{(k)}} \cdot \frac{1}{\sqrt{Var[x^{(k)}] + \epsilon}}, & \text{otherwise} \end{cases} \quad , \quad (6)$$

while the others remain the same format. Here $S$ is the location set for sampled neurons. For neurons outside $S$, they didn't participate in the estimation of the mean and variance in the forward pass, so $\frac{\partial E[x^{(k)}]}{\partial x_i^{(k)}}$ and $\frac{\partial Var[x^{(k)}]}{\partial x_i^{(k)}}$ equal to zeros. We should note that although we sampled only $s$ data in the forward pass, the reduction operations in the backward pass still have $m$ length since the neurons outside $S$ also participate in the activation normalization using the estimated statistics from the sampled data. Therefore, there is no theoretical speedup for the backward pass. In our experiments, to assemble a dense and efficient addition operation with $\frac{\partial l}{\partial \widehat{x}_i^{(k)}} \cdot \frac{1}{\sqrt{Var[x^{(k)}] + \epsilon}}$ for all neurons, the mentioned zeros outside $S$ are pre-padded. However, it has great potential for memory space and access reduction since the $\frac{\partial E[x^{(k)}]}{\partial x_i^{(k)}}$ and $\frac{\partial Var[x^{(k)}]}{\partial x_i^{(k)}}$ outside $S$ are zeros, which can be leveraged by specialized devices.

**Reason for only Speeding up Forward Pass**. As shown in equation (1) & (5), although the forward pass only takes $\frac{1}{3}$ reduction operations in BN, it requires nearly $\frac{2}{3}$ time. The underlying reason is that the reduction operations in the backward pass can be parallelized while the second reduction operation in the forward pass used to calculate variance cannot operate until the mean is provided.

**Regular Execution Pattern**. To approach the theoretical speedup, we suggest using more regular sampling pattern such as the continuous samples in BS and intra-FM rectangular shape and inter-FM shared location in FS. A regular sampling pattern can improve the cache usage by avoiding random access. Moreover, the operation in the backward pass corresponding to the sampling operation in the forward pass is "padding" shown in Fig. 1(d). Under a regular sampling pattern, the padding is block-wise that becomes much easier. On the other side, intuitively, a static computational graph can be calculated faster and easier deployed on various platforms for (1) the static graph's pipeline can be optimized once for all before execution, (2) popular deep learning frameworks like TensorFlow (Girija, 2016) are developed as a static computational graph. For these reasons, we expect the random sampling is achieved through a static computational graph, which is obtained by updating the sampling indexes only per training epoch.

## APPENDIX D    INFLUENCE OF DECAY RATE FOR MOVING AVERAGE

During each validation step after certain training iterations, $E[x^{(k)}]$ & $Var[x^{(k)}]$ are replaced with the recorded moving average, which is governed by

$$X_{ma}[x^{(k)}]_{it} = \begin{cases} X[x^{(k)}]_{it}, it = 1 \\ \alpha X[x^{(k)}]_{it} + (1-\alpha)X_{ma}[x^{(k)}]_{it-1}, it \in [2, \frac{N}{B}] \end{cases}, \tag{7}$$

where $X$ stands for "$E$" or "$Var$", $X_{ma}[x^{(k)}]$ denotes the moving average, $it$ is the iteration number, and $\alpha$ is the decay rate. Based on equation (7), wherein the variance of $X_{ma}[x^{(k)}]_{it}$ equals to $\alpha^2 Var[X[x^{(k)}]_{it}] + (1-\alpha)^2 Var[X_{ma}[x^{(k)}]_{it-1}]$, if we assume that (1) each estimated value shares the same variance $Var[X[x^{(k)}]_{it}]$, (2) they are independent with each other, and (3) the iteration number goes sufficiently large, we will get

$$Var[X_{ma}[x^{(k)}]_{it}] \approx \frac{\alpha}{2-\alpha} Var[X[x^{(k)}]_{it}]. \tag{8}$$

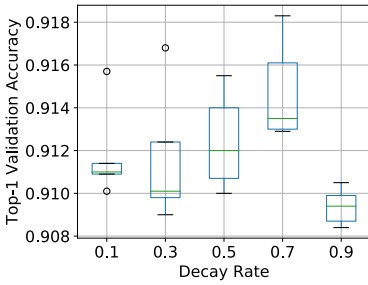

Figure 11: Influence of decay rate for moving average.

The above equation reveals that an appropriately smaller $\alpha$ might scale down the estimation error, thus produces better validation accuracy. To verify this prediction, the experiments are conducted on ResNet-56 over CIFAR-10 and using BS-1/128(0.78%) sampling. As shown in Fig. 11, it's obvious that there exists a best decay rate setting (here is 0.7) whereas the popular decay rate is 0.9. The performance also decays when decay rate is smaller than 0.7, which is because a too small $\alpha$ may lose the capability to record the moving estimation, thus degrade the validation accuracy. This is interesting because the decay rate is usually ignored by researchers, but the default value is probably not the best setting in our context.

## APPENDIX E    REDUCTION OPERATIONS IN BN

To further demonstrate that reduction operations are the major bottleneck of BN, we build a pure BN network for profiling as shown in Fig. 12. The network's input is a variable with shape of [128, 112, 112, 64] which is initialized following a standard normal distribution. The network is trained for 100 iterations and the training time is recorded. We overwrite the original BN's codes and remove the reduction operations in both forward pass and backward pass for contrast test. We use three different GPUs: K80, Titan Xp, and Tesla V100 to improve the reliability. The results are shown in Table 5. We can see that on all the three GPUs, reduction operations take up to >60% of the entire operation time in BN. As a result, it's solid to argue that the reduction operations are the bottleneck of BN.

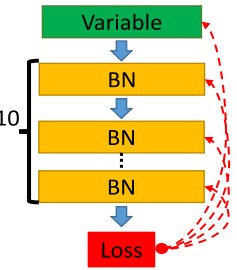

Figure 12: BN network for profiling.

Table 5: The execution time in different ablation study of reduction operations in BN.

| GPU | BN | w/o Reduction | w/o BN | Reduction Percentage |
|---|---|---|---|---|
| K80 | 107.32s | 49.49s | 9.96s | 59.4% |
| Titan XP | 64.27s | 29.04s | 9.80s | 64.7% |
| Tesla V100 | 21.82s | 13.14s | 7.86s | 62.2% |

## APPENDIX F    MICRO-BN EXTENSION

Micro-BN aims to alleviate the diminishing of BN's effectiveness when the amount of data in each GPU node is too small to provide a reliable estimation of activation statistics. Previous work can be classified to two categories: (1) Sync-BN (Zhang et al., 2018a) and (2) Local-BN (Ba et al., 2016; Wu & He, 2018; Ioffe, 2017; Wang et al., 2018). The former addresses this problem by synchronizing the estimations from different GPUs at each layer, which induces significant inter-GPU data dependency and slows down training process. The latter solves this problem by either avoiding the use of batch dimension in the batched activation tensor for statistics or using additional information beyond current layer to calibrate the statistics.

In terms of BN's efficiency preservation, we are facing a similar problem with micro-BN, thus our framework can be further extended to the micro-BN scenario. In Sync-BN: (1) **With FS**, each GPU node executes the same patch sampling as normal FS; (2) **With BS**, we can randomly select the statistics from a fraction of GPUs rather than all nodes; (3) **With VDN**, the virtual samples can be fed into a single or few GPUs. The first one can just simplify the computational cost within each GPU, while the last two further optimize the inter-GPU data dependency. In Local-BN, since the available data for each GPU is already tiny, the BN sampling strategy will be invalid. Fortunately, the VDN can still be effective by feeding virtue samples into each node.

**Experiments**. The normalization in Sync-BN is based on the statistics from multiple nodes through synchronization, which is equivalent to that in Fig. 6 with large batch size for each node. Therefore, to avoid repetition, here we just show the results on Local-BN with VDN optimization. We let the overall batch size of 256 breaks down to 64 workers (each one has only 4 samples for local normalization). We use "(gradient batch size, statistics batch size)" of (256, 4) to denote the configuration (Wang et al., 2018). A baseline of (256, 32) with BN and one previous work Group Normalization (GN) (Wu & He, 2018) are used for comparison. As shown in Fig. 13, although the reduction of batch size will degrade the model accuracy, our VDN can achieve slightly better result (top-1 validation error rate: 30.88%) than GN (top-1 validation error rate: 30.96%), an advanced technique for this scenario with tiny batch size. This promises the correct training of very large model so that each single GPU node can only accommodate several samples.

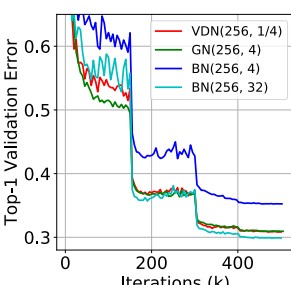

Figure 13: Training curves in micro-BN scenario. (ImageNet, ResNet 18)

## APPENDIX G    ORGANIZATION OF THE WHOLE PAPER

.

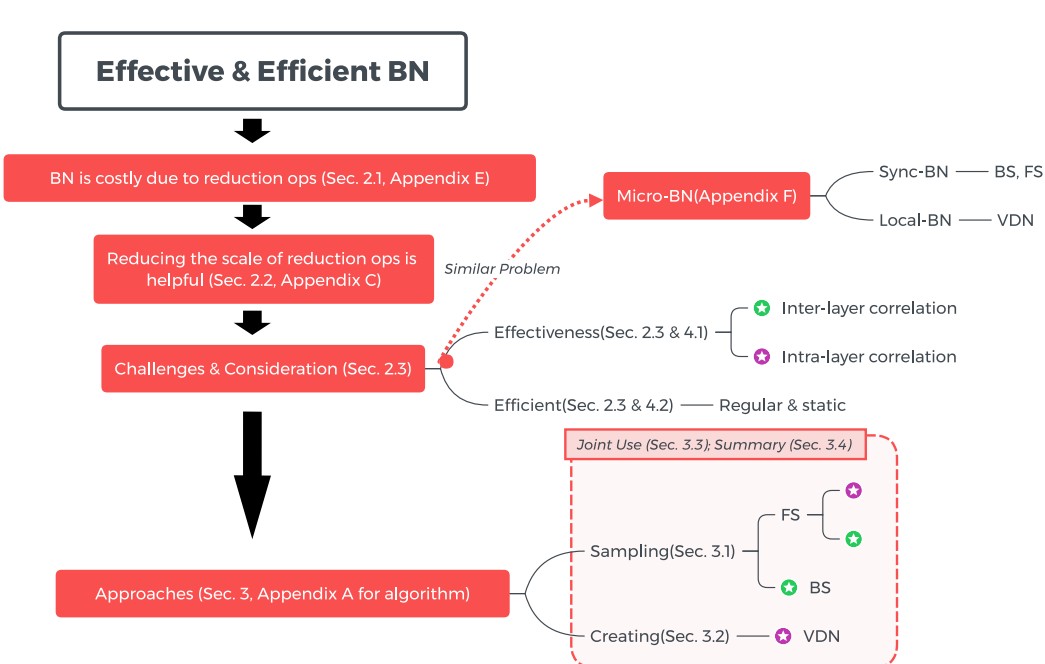

Figure 14: Illustration of the paper's organization. The Green and Purple markers (round circle with a star in the center) represent whether the effectiveness is preserved by reducing inter-layer or intra-layer correlation (Green: inter-layer; Purple: intra-layer). Moreover, the consideration of regular & static execution pattern is applied to all approaches.

