# OpenReview forum: "Effective and Efficient Batch Normalization Using Few Uncorrelated Data for Statistics' Estimation"
_ICLR.cc/2019/Conference_

### Official Review · AnonReviewer3 · 2018-10-31
**limited improvements**

**Rating:** 5
**Confidence:** 5

**Review:**

This paper proposes to use subsampling to reduce the computation cost of BN, which buys around 20% of the computational cost.
- In normal BN, the gradient is propagated through the normalization factor as well, how would that change in the case of subsampled BN?
- The minimum amount of gains makes it less appealing considering the potential complexity of implementing the algorithm.
- Did the author compared against cuDNN’s native version of BN? Because random sampling is involved, this will result in less regular patterns of computation, this could likely make the implementation of BN to be less efficient.
In summary, the method proposed in the paper is reasonable but could be limited in practice due to only 20% maximum gain can be achieved.

Update after the author response:

The author addressed some of the concerns raised in the review(Thanks for the detailed response), in particular, the comparison to cuDNN.  I still think the paper is still borderline but the results might be of interest to some of the ICLR audience.

---

> ### Author Response · Authors · 2018-11-21
> **Responses to Reviewer 3**
>
>
> Thanks for your constructive feedback. In order to make our paper more understandable, we add a figure that illustrates the organization of the whole paper in Appendix G.
>
> (1)	Gradient Propagation:
>
> The influence of BN sampling on the gradient propagation of activations has been provided in the previous submission (Equation (5)-(6), Appendix C). The gradient expressions of the normalization parameters (i.e. gamma and beta, mean and variance) remain the same as the vanilla BN (equation (5), Appendix C). The only change of backward pass happens when calculating the gradient of the input (equation (6), Appendix C), where the gradients of mean & variance are multiplied with the ‘s’ (sampling ratio)-dependent variables or padded with zeros. This is also illustrated in Fig. 1(d).
>
> (2)	20% Speedup Gains:
>
> ~20% (actually up to 21.7%) speedup seems not enough, but we must note that it is an overall training speedup practically achieved on modern GPUs (Titan XP and Tesla V100) including all the necessary overheads. The challenges can be summarized as follows:
>
> a)	To achieve acceleration on GPUs without the support of specialized libraries is a tough task. Many of previous work on DNN compression (e.g. quantization or sparsification) cannot obtain practical acceleration on GPUs even if they claimed very high compression ratio or they have the potential to achieve speedup on specialized platforms.
>
> b)	To achieve overall training speedup on GPUs is more difficult. Although many of the previous work presented acceleration, they focused on the inference phase. They usually made the training more complicated due to various regularization constraints and iterative retraining.
>
> To the best of our knowledge, we didn’t see publication achieving >20% overall training acceleration practically achieved on modern GPUs except for the case of distributed training. In distributed training, the bottleneck is on the inter-GPU communication cost (e.g. weight gradients) which can be greatly improved by compressing the communication data. Our solution focuses on simplifying the intra-GPU computation, which is orthogonal to those methods.
>
> (3)	cuDNN Comparison:
>
> We did compare against cuDNN’s native version of BN. Our speedup was practically achieved on GPU platforms rather than theoretical estimation, which includes all the necessary overheads. The comparison baseline was vanilla DNN training on GPUs with cuDNN support, and the implementation of our methods was described in Algorithm 1-3, Appendix A.
>
> Yes, to reduce the overhead caused by random sampling is indeed a challenge for practical acceleration, which expects more regular execution pattern. In reality, the situation is more complicated because we also need more randomness to reduce the correlation between sampled data for better statistics estimation. Therefore, the estimation accuracy and the execution acceleration form a trade-off, which is the major motivation of this paper to win more gains on both sides. Our solutions are summarized as below:
>
> In BN sampling, in order to achieve accurate statistics, the sampling indexes are independent across layers to reduce the data correlation. Specifically, BS selects different samples across layers while FS selects different patch locations (within feature maps) across layers. This indicates the sampling is non-uniform in the layer dimension which can prevent the estimation error scaling up along the network depth. These sampling considerations enable us to achieve a much lower sampling ratio with negligible accuracy loss. In the meantime, to achieve regular execution pattern for practical acceleration, 1) in BS the sampling indexes are continuous across different inputs within a mini-batch and are shared by different channels; 2) in FS the sampling patch shape is rectangular and the patch location is shared by different channels. The random indexes are updated only once for each epoch rather than each iteration to improve the execution efficiency via static computation graph during each epoch.
>
> All of the above design considerations and sampling strategies have already been provided in Section 3.3 and Section 4.1, respectively, in the original submission. In the revised version, we have tried our best to make them clearer in Section 2 & 3 of the revised version.

---

### Official Review · AnonReviewer2 · 2018-11-02

**Rating:** 5
**Confidence:** 3

**Review:**

This paper proposes a new technique that can reduce the computational complexity of batch normalization. Several sampling methods called NS, BS, and FS are proposed, and additionally, VDN is proposed to generate random virtual samples.  Experiment results follow to support the authors' goal.

pros)
(+) The paper is clearly written and easy to follow.
(+) The way of reducing the computational cost looks good.
(+) The method can be easily adapted to BN or other batch-based methods.

cons)
(-) Any motivations or insights into NS, BS, and FS are not provided. Furthermore, the proposed sampling strategy looks heuristic without any studies.
(-) For VDN, how to generate virtual samples is not clearly stated. I think the way of generating samples is critical to the performance of VDN but hard to find the exact way to do that.
(-) How to determine the sampling ratio for each normalization method is not provided, and it would be better if the authors can show some studies about sampling ratio versus the speed gain.
(-) It is hard to choose which normalization among FS and BS is better as looking at Table 2 and 3 only. So how about the speedup using BS+VDN?

comments)
- It is something strange why the authors used shallower ResNet on ImageNet and deeper ones on CIFAR datasets, maybe it was due to the training time, but the authors should clarify it.
- What is the goal of the correlation analysis section? Especially, Figure 7 looks similar among BS, FS, VDN, and NS. Furthermore, the authors could include BN into the comparison.
- This kind of paper should incoporate different ablation studies as much as the authors can, but it seems to be lacking.


The paper has an interesting idea about sampling some features to speed up the batch normalization. However, it looks quite obvious and needs more experimental grounds such as ablation studies to support the idea.

---

> ### Author Response · Authors · 2018-11-21
> **Responses to Reviewer 2 (Part I)**
>
>
> Thanks for your insightful comments. In order to make our paper more understandable, we add a figure that illustrates the organization of the whole paper in Appendix G. Details can be found as below.
>
> (1)	Motivation and Insights:
>
> a)	Issue: BN is costly in DNN training (see Fig. 1a).
>
> b)	Opportunity: We propose using just a small fraction of data to do BN efficiently.
>
> c)	Challenge: The key challenge is to achieve both more accurate statistics to preserve the BN’s effectiveness for reducing accuracy loss and more regular execution pattern for practical acceleration. For the former, the correlation between sampled data need to be as less as possible; while for the latter, more regular execution pattern is expected. This is a difficult trade-off that demands careful design.
>         As shown in Fig. 4, the accuracy of NS (a naïve sampling strategy as the baseline) drops drastically as the sampling ratio decreases, which indicates simple heuristic sampling cannot work very well. The accuracy loss is caused by two factors: 1) within each layer, the estimation variance increases high when the sampling ratio is low; 2) the mean & variance are easy to scale up as network deepens (see Fig. 2).
>
> d)	Approach (Design Insights): In BN sampling, in order to achieve accurate statistics, the sampling indexes are independent across layers to reduce the data correlation. Specifically, BS selects different samples across layers while FS selects different patch locations (within feature maps) across layers. This indicates the sampling is non-uniform in the layer dimension which can prevent the estimation error scaling up along the network depth. In the meantime, to achieve regular execution pattern for practical acceleration, 1) in BS the sampling indexes are continuous across different inputs within a mini-batch and are shared by different channels; 2) in FS the sampling patch shape is rectangular and the patch location is shared by different channels. The random indexes are updated only once for each epoch rather than each iteration to improve the execution efficiency via static computation graph during each epoch.
>         In VDN, the major design philosophy is to create uncorrelated data for producing extremely less virtual samples with good global statistics. The significance of VDN lays not in how it is performed but the philosophy behind it. In VDN, the data within virtual samples are generated i.i.d (independent and identically distributed). The independence between them ensures a reliable estimation with much fewer data.
>
> In general, we acknowledge that the challenges of using few data to conduct BN (i.e. guarantee both accurate statistics and practical acceleration) and how our methods (i.e. BN sampling and VDN) solve them effectively and efficiently (i.e. design philosophies) were not presented very well in the previous submission. Now, we have re-written the Introduction (Section 1) and the Design Philosophies (originally in Section 3.3, now it is moved to Section 2) to make our motivation and contribution clearer. We modified the paper title to be “Effective and Efficient Batch Normalization using Few Uncorrelated Data for Statistics Estimation” for better readability. The new title can reflect that our motivation is to do BN more efficiently by using just few data, in which carefully designed strategies for reducing the data correlation are proposed to guarantee the BN effectiveness.
>
> (2)	VDN Implementation:
>   The virtual samples in VDN are tensors wherein each element simply follows N(μ, σ). μ and σ are the statistics of the whole training dataset, which are calculated offline. All tensor elements are i.i.d..The implementation of VDN was clearly presented in Algorithm 3 of the previous Appendix A. We have made it clearer in the main text (Section 3.2, footnote 3) of the revised manuscript.
>
> (3)	Sampling Ratio Determination:
>
> A theoretical way to determine sampling ratio is hard since it is difficult to evaluate how much correlation we can reduce. The whole process is a trade-off between accuracy and efficiency and our methods help win more gains on both sides. Our experiments illustrate that on large datasets and deep networks (e.g. ImageNet, Densenet-121), the accuracy loss is still negligible (-0.6%) even if under a very low sampling rate of 4.7% (Table 3). Therefore, we believe it is safe to choose a relatively low sampling ratio.
>
> For the suggested study of “sampling ratio v.s. speed gain”, we have added additional experiments in the revised manuscript, which can be found in newly added Figure 10 and the revised Table 4.
>
> To be continued in Part II...

---

> > ### Author Response · Authors · 2018-11-21
> > **Responses to Reviewer 2 (Part II)**
> >
> >
> > (4)	Method Selection:
> >
> > There is actually no absolute answer for “FS & BS, which is better”. FS is more capable of maintaining the accuracy, convergence rate, and is more stable under extremely low sampling ratio. However, its execution pattern is less regular than BS since it samples data within each sample, which can explain why it provides lower acceleration than BS.
> >
> > The speedup of 23.8% when using “BS+VDN” is definitely higher than the 21.7% of “FS+VDN” on DenseNet-121, whereas it will lose more accuracy. We have added its results into the revised Table 3-4.
> >
> > (5)	Network Depth:
> >
> > Yes, it is truly due to the training time, and we will clarify this point in the revised section of Experimental Setup. However, we have already evaluated our method on deeper networks over ImageNet (e.g. DenseNet-121 in Table 3-4 and Fig. 6) in the initial submission to demonstrate the effectiveness of our methods.
> >
> > (6)	The goal of Correlation Analysis:
> >
> > The goal of section “Correlation Analysis” is to demonstrate that the proposed methods can effectively reduce the data correlation, which is one of the major design insights as aforementioned. The difference between the different methods in Fig. 7 can be seen from the centroid of errors recorded by the triangles. Since the plot is in Fig. 7 is in logarithmic scale, the differences will be shrunk.
> >
> > The reason that the BN was not included is that we use its result as the ground-truth value to be estimated, based on which the estimation error is calculated.
> >
> > (7)	Ablation Study:
> >
> > Actually the “correlation analysis” also acts as an ablation study for studying the role of data correlation towards accurate statistics estimation and Fig. 9(c) is also an ablation study for studying the role of regular execution pattern towards practical acceleration.
> >
> > 1.	"inter-layer correlation": the only difference between NS and BS is that BS has a much lower inter-layer correlation as shown in Fig. 8, since BS selects samples within each mini-batch independently for each layer. Therefore, BS can achieve much better accuracy compared to NS under the same sampling ratio.
> >
> > 2.	“intra-layer correlation”: Also as shown in Fig. 8, although FS has higher inter-layer correlation compared with BS since FS involves all samples in each mini-batch (just samples a fraction of data within each sample), it still outperforms BS because of the lower intra-layer correlation (see Fig. 7, FS has much lower estimation error). VDN can achieve comparable estimation error compared to NS but using much fewer data for its data are independent with each other.
> >
> > 3.	“regular execution patter”: From Fig. 9(c) we can see that the practical speedup is much lower without regular execution pattern (including regular sampling shape and static sampling index during each training epoch).

---

### Official Review · AnonReviewer1 · 2018-11-06
**Good effort, but has not demonstrated enough evidence to convince me to accept it**

**Rating:** 4
**Confidence:** 3

**Review:**

The paper proposes a sampling-based method that aims at accelerating Batch Normalization (BN) during training of a neural network.

Quality:
   The writing of the paper needs more polishing; I saw grammatical errors here and there: for example, at the first paragraph of page 2, "alternating" should be alternative and "synthetical" should be synthetic...

Clarity:
  I have not been able to fully understand why the proposed (uniform) sampling variant of BN is better than previous effort at making BN less computationally expensive in a GPU-based training environment by reading the paper:
  1. The authors argue that the "summation" operation is the one that makes BN expensive; however, the authors have not demonstrated enough evidence of this argument
  2. If "summation" operation is what makes BN expensive, then in a GPU-based environment, can we simply divide the data into smaller batch, and train on each GPU using a smaller batch (this is way, each GPU is essentially calculating the statistics based on a sub-sample)
  3. The authors discussed Micro-BN, which "alleviate the diminishing of BN's effectiveness when the amount of data in each GPU node is too small..." This seems to show that in practice, training with BN does not suffer from having a large batch, but instead suffers from having too small batch size on each GPU node. Related to the point above, doesn't this observation play against the motivation of using a sampling-based BN?

Originality and Significance:
  I think the effort of trying to make BN less computationally heavy is respectable. But the idea of uniform-sampling seems     rather straight-forward, and more important, I do not see its justification from reading the paper; the other technique introduced, Virtual Dataset Normalization, seems to be a direct application of Virtual Batch Normalization (Salimans et al 16).

---

> ### Author Response · Authors · 2018-11-21
> **Responses to Reviewer 1 (Part I)**
>
> We sincerely appreciate your valuable comments. In order to make our paper more understandable, we add a figure that illustrates the organization of the whole paper in Appendix G. Our responses are listed as follows.
>
> (1)	The summation is the Bottleneck:
>
> Because “summation” requires reduction operation to aggregate many data, the data movement becomes the bottleneck. Other operations in BN belong to element-wise arithmetics, which can be completed without much costly data movement. Moreover, even with parallel adder tree, the reduction operations cannot fully exploit the parallel capacity of GPUs due to the exponentially decreasing adders as tree depth increases. In the revised version (Appendix E), we added experiments to evidence the BN’s bottleneck. It can be seen that “summation” occupies over 60% operation time.
>
> (2)	Use Small Batch on Each GPU:
>
> Simply dividing the data into smaller batches and training one certain batch on each GPU will cause two problems:
>
> a)	First, each GPU will be underutilized since it only processes a smaller batch, and we require more GPU resources, which also results in more communication cost. This is not practical and not efficient in reality.
>
> b)	Second, each GPU only use the local statistics of a smaller batch, which will degrade the convergence of SGD learning. The reason is that when the amount of data used for statistics is small, the variance of estimated statistics of the whole dataset would grow and even be unreliable. This is the well-known problem of micro-BN.
>
> In real scenarios, we usually configure the batch size to utilize the resources as full as possible. Then the batch size for each GPU is usually determined by the model size.
>
> (3)	Does “BN Suffers from Small Batch” Play against “BN Sampling”:
>
> The small amount of data usually degrades BN’s effectiveness through the variance scaling of the estimated mean & variance. However, our motivation of using a sampling-based BN does not play against the observation of micro-BN, because we can address the above issue by carefully designing the sampling strategies rather than using a straightforward way. This is just the major contribution of our work.
>
> As a result, although with only a small fraction of data, less data correlation can produce much lower estimation variance than using correlated data (especially positively correlated data), which will make the estimation of global statistics more accurate. Therefore, through sampling uncorrelated data (BS & FS) or directly creating uncorrelated data (VDN), we can still preserve BN’s effectiveness even if just using few data.
>
> Thanks so much for your question that can really help improve our paper quality. The previous tile was ‘Batch Normalization Sampling’ which might not be able to present our motivation and design insights very well. In the revised version, we modify the tile to be “Effective and Efficient Batch Normalization using Few Uncorrelated Data for Statistics Estimation”. This reflects that our motivation is to do BN more efficiently by using just few data, in which carefully designed strategies for reducing the data correlation are proposed to guarantee the BN effectiveness. Now, we incorporate the BS & FS (sampling uncorrelated data) and VND (creating uncorrelated data) into a unified framework of how to do BN effectively and efficiently in the meantime. The strategies to simultaneously reduce accuracy loss and achieve practical acceleration will be discussed in the following answers.
>
> To be continued in Part II...

---

> > ### Author Response · Authors · 2018-11-21
> > **Responses to Reviewer 1 (Part II)**
> >
> >
> > (4)	Originality and Significance (not uniform sampling & not naive virtual BN):
> >
> > It has to be clarified that our approaches (BS, FS, VDN) are not straightforward sampling but carefully designed based on two considerations towards the goal of just using few data to do BN well:
> >
> > a)	BN’s effectiveness should be preserved by 1) reducing the estimation error within each layer (The direct effect of sampling is the increased estimation variance, which can be solved by selecting less correlated data within each layer in our approaches.) and 2) preventing the mean & variance from scaling up as network deepens.
> >
> > b)	The execution pattern should be regular enough for GPU to achieve practical acceleration.
> > The above two considerations form a difficult trade-off that demands careful design. Of course, a straightforward sampling (e.g. NS-naïve sampling and FRS-fully random sampling) would cause more accuracy loss (see Fig. 4-6; Tab. 3). This is just the major contribution of our work to guarantee the BN effectiveness using just few data for statistics estimation as well as improve the execution efficiency in the meantime, Specifically.
> >
> > a)	For BN Sampling:
> >
> > To achieve accurate statistics, the sampling indexes are independent across layers to reduce the data correlation which indicates the sampling is non-uniform in the layer dimension. Specifically, BS selects different samples across layers while FS selects different patch locations (within feature maps) across layers. This can prevent the estimation error scaling up as network deepens. In the meantime, to achieve regular execution pattern for practical acceleration, 1) in BS the sampling indexes are continuous across different inputs within a mini-batch and are shared by different channels; 2) in FS the sampling patch shape is rectangular and the patch location is shared by different channels. The random indexes are updated only once for each epoch rather than each iteration to improve the execution efficiency via static computation graph during each epoch.
> >
> > b)	For VDN:
> >
> > The major philosophy for VDN is to create uncorrelated data for producing extremely less virtual samples with good global statistics. The difference between VDN and previous VBN (Salimans et al. 2016) has been discussed in our original submission (Section 4.2). The significance of VDN lays not in how it is performed but the philosophy behind it. In VDN, the data within virtual samples are generated i.i.d (independent and identically distributed). The independence between them ensures a reliable estimation with much fewer data. Whereas VBN has a totally different philosophy: it is introduced to reduce the dependence between input samples rather than creating new independent data in the generator networks. This can justify the originality of VDN. We will clarify this with more details in the revised Section 3.2.
> >
> > In general, we acknowledge that the challenges of using few data to conduct BN (i.e. guarantee both accurate statistics and practical acceleration) and how our methods (i.e. BN sampling and VDN) solve them effectively and efficiently (i.e. design philosophies) were not presented very well in the previous submission. Now, we have re-written the Introduction (Section 1) and the Design Philosophies (originally in Section 3.3, now it is moved to Section 2.) to make our motivation and contribution clearer. As aforementioned, we modified the paper title to be “Effective and Efficient Batch Normalization using Few Uncorrelated Data for Statistics' Estimation” for better readability.
> >
> > (5)	Grammatical Errors:
> >
> > We have tried our best to polish them in the revised manuscript.

---

### Meta-Review · Area_Chair1 · 2018-12-09
**a reasonable proposal for speeding up batch norm, but somewhat obvious and with limited practical benefit**

**Confidence:** 5
**Recommendation:** Reject

**Metareview:**

This paper proposes a faster approximation to batch norm, which avoids summing over the entire batch by subsampling either random examples or random image locations. It analyzes some of the tradeoffs of computation time vs. statistical efficiency of gradient estimation, and proposes schemes for decorrelating the samples to make good use of smaller numbers of samples.

The proposal is a reasonable one, and seems to give a noticeable improvement in efficiency. However, it's not clear there is a substantial enough contribution for an ICLR paper. The idea of subsampling is fairly obvious, and various other methods have already been proposed which decouple the computation of BN statistics from the training batch. From a practical standpoint, it's not clear that the observed benefit is large enough to justify the considerable complexity of an efficient implementation.